# Cryogenic characteristics of graphene composites—evolution from thermal conductors to thermal insulators

Zahra Ebrahim Nataj [1,4], Youming Xu[2,4], Dylan Wright[1], Jonas O. Brown [1], Jivtesh Garg[3], Xi Chen [2], Fariborz Kargar [1] ✉ & Alexander A. Balandin [1] ✉

The development of cryogenic semiconductor electronics and superconducting quantum computing requires composite materials that can provide both thermal conduction and thermal insulation. We demonstrated that at cryogenic temperatures, the thermal conductivity of graphene composites can be both higher and lower than that of the reference pristine epoxy, depending on the graphene filler loading and temperature. There exists a well-defined cross-over temperature—above it, the thermal conductivity of composites increases with the addition of graphene; below it, the thermal conductivity decreases with the addition of graphene. The counter-intuitive trend was explained by the specificity of heat conduction at low temperatures: graphene fillers can serve as, both, the scattering centers for phonons in the matrix material and as the conduits of heat. We offer a physical model that explains the experimental trends by the increasing effect of the thermal boundary resistance at cryogenic temperatures and the anomalous thermal percolation threshold, which becomes temperature dependent. The obtained results suggest the possibility of using graphene composites for, both, removing the heat and thermally insulating components at cryogenic temperatures—a capability important for quantum computing and cryogenically cooled conventional electronics.

There is a rapidly emerging need for thermal management at cryogenic temperatures. It is driven by several trends and developments. There is a strong motivation to run conventional semiconductor electronics at low temperatures to implement "cold computing," which allows one to increase computational and energy efficiency while reducing power consumption[1,2]. The main progress with quantum computing technologies is associated with superconducting qubits, which require cryogenic temperatures[3]. Space exploration needs electronics operating in harsh low-temperature environments. Cryogenic thermal management usually relies on both thermal conductors and thermal insulators[4–6]. The former is the polymer-based thermal interface materials (TIMs) with fillers that conduct heat well, facilitating heat removal, and the latter are polymer materials, which have low thermal conductivity and can act as thermal insulators between electronic components operating at different temperatures. Examples of low-temperature thermal management applications include protective coatings of superconductive power cables[7], adhesives in cryogenic low-noise amplifiers for radio-astronomy and space communication systems[8], optical mounts of cryogenic refractive optics, and cryosorption pumps[9,10].

[1]Phonon Optimized Engineered Materials Center, Department of Electrical and Computer Engineering, University of California, Riverside, CA 92521, USA. [2]Department of Electrical and Computer Engineering, University of California, Riverside, CA 92521, USA. [3]Department of Aerospace and Mechanical Engineering, University of Oklahoma, Norman, OK 73019, USA. [4]These authors contributed equally. Zahra Ebrahim Nataj, Youming Xu. ✉ e-mail: fkargar@ece.ucr.edu; balandin@ece.ucr.edu

Polymers are poor heat conductors with thermal conductivity in the range from ~0.2 to 0.5 Wm$^{-1}$K$^{-1}$ at room temperature (RT)[11,12]. Polymers are used in TIMs as the base, i.e., matrix material, that fills the air gaps between two adjacent solid interfaces and provides adhesive functionality when used in curing composites. The common strategy to increase the thermal conductivity of polymers is to add micrometer- and nanometer-scale fillers with a higher intrinsic thermal conductivity that can couple well with the base polymer. A mixture of the single-layer and few-layer graphene (FLG) flakes, termed "graphene" in the thermal context, has proven to be an efficient filler material for a variety of TIMs, including non-curing mineral oil-based thermal pastes[13], and curing epoxies[12,14,15]. Graphene for thermal management applications can be mass-produced via liquid-phase exfoliation, graphene oxide reduction, or other techniques[16,17]. Graphene TIMs with thermal conductivity above ~12 Wm$^{-1}$K$^{-1}$ near RT, which exceeds the metric of conventional commercial TIMs, have been reported by several research groups[12,14,15]. The excellent performance of graphene TIMs near RT originates in the extraordinarily high intrinsic thermal conductivity of graphene and few-layer graphene[18,19], strong coupling to the matrix, good dispersion, and appropriate viscosity range of the resulting composites[20]. The enhancement of thermal conductivity is achieved both below and above the thermal percolation threshold—a loading fraction at which the graphene fillers start to form continuous thermally conductive paths[21,22]. The thermal percolation threshold can be identified when the dependence of the thermal conductivity on filler loading becomes super-linear[21,23]. One should note that the thermal properties of graphene composites have only been studied at RT and above—the temperature range of interest for conventional electronics. We are not aware of any report on the cryogenic thermal characteristics of graphene composites. In general, the data for the thermal properties of any polymer composites at cryogenic temperatures are scarce. The understanding of heat propagation in amorphous polymers at low temperatures is far from complete, even if one does not consider the solid inclusions, i.e., fillers[24,25].

Here, we investigated the thermal properties of epoxy–graphene composites at temperatures from 2 K to RT. Epoxy is a practically important material and is often used as a reference material to compare the effect of different types of fillers on its thermal conductivity. We found that at cryogenic temperatures, the thermal conductivity of graphene composites can be both higher and lower than that of the reference pristine epoxy, depending on the graphene filler loading and temperature. This is drastically different from what is observed near RT. Moreover, there exists a well-defined cross-over temperature that above it, the thermal conductivity increases with the addition of graphene, whereas, below it, the thermal conductivity decreases with the addition of graphene. Graphene composites are unique in a way that they can provide both the strongest enhancement in thermal conductivity and the strongest suppression. We offer a physical model explanation of the counter-intuitive trends and provide numerical simulation data which agree with the measurements. The obtained results suggest the possibility of using composites with the same constituent materials for, both, removing the heat and thermally insulating electronic components at cryogenic temperatures. The latter constitutes a *conceptual* change for thermal management, which typically rely on different materials for heat conduction and isolation.

## Results

### Materials

The polymer matrix is a thermoset epoxy set consisting of a base resin (Bisphenol-A, Allied HighTech Products, Inc.) and a hardening agent (Triethylenetetramine; Allied HighTech Products, Inc.). We used few-layer graphene with the specified average lateral dimension of 25 μm, an average thickness of 15 nm, and an average surface area of 50 to 80 m$^2$g$^{-1}$ (xGnP, Grade H, XG Sciences, the US) as the fillers for the preparation of the composites. The graphene fillers were subjected to

further exfoliation during the high-speed mixing process with the base polymer matrix. The resulting fillers constitute a mixture of FLG and single-layer graphene. The lateral dimensions of graphene fillers are an important parameter for tuning the thermal conductivity of the composites[26]. To maximize the thermal conductivity of the composite, one normally wants to keep the lateral dimensions above the gray phonon mean free path (MFP) in graphene, which is ~1 μm near RT[27]. In contrast, to improve the insulating properties of composites, fillers with smaller lateral dimensions are preferred[28].

### Composite preparation and characterization

Several composite samples were prepared by mixing precalculated quantities of the resin, the hardener, and the FLG fillers to hit a targeted filler loading level. First, a certain amount of FLG fillers was distributed into the epoxy resin using a high-speed shear mixer (Flacktek, Inc., the USA). The hardener was then added to the mixture in the mass ratio of 12:100 with respect to the base epoxy resin's weight. The blend was vacuumed several times to eliminate any air bubbles that might have been trapped during the composite preparation process. The mixture was poured into special molds to cure and solidify (see Supplementary Fig. 1 for the optical images of the samples). The results of the mass density measurements confirm that the porosity of the samples is negligible (Supplementary Fig. 2). The details of the composite preparation can be found in the Supplemental Information. Optical and scanning electron microscopy (SEM) images of two representative samples with graphene loadings of 2.6 vol% and 18.0 vol% are presented in Fig. 1a–f, respectively. A few fillers in the SEM images are shown in different colors than that of the background epoxy host to illustrate the randomness of the filler distribution in the polymer matrix. It should be noted that as the volume fraction of fillers increases beyond a certain loading, referred to as the percolation threshold, the fillers start to overlap (see the SEM image in Supplementary Fig. 3). At and beyond the percolation regime, fillers create a network of electrically and thermally conductive pathways within the base polymer matrix[21,22,29]. The percolation results in significant enhancements in both the electrical and thermal characteristics of the composites[21,22,29].

The composites were further characterized using Raman and Brillouin–Mandelstam light scattering spectroscopy, also referred to as Brillouin light scattering spectroscopy[30–32]. The Raman spectroscopy (Renishaw InVia) was carried out under laser excitation with the wavelength of $\lambda = 633$ nm in the conventional backscattering configuration. In all experiments, the laser power was kept at ~3 mW. Raman spectra of several samples at random spots are presented in Fig. 1g. In all composites, the characteristic G-peak and 2D peak of graphene were observed at ~1580 cm$^{-1}$ and ~2700 cm$^{-1}$ [33], confirming an even distribution of graphene throughout the samples. The intensity of the 2D peak is much lower than the G-peak indicating the presence of a mixture of single-layer and few-layer graphene in the composite samples. The Brillouin spectroscopy allows probing low-energy acoustic phonons near the Brillouin-zone (BZ) center with energies in the range of 2 GHz to 900 GHz[30,34–36]. In bulk polymers, the Brillouin spectrum is dominated by the inelastic scattering of light from bulk phonons, i.e., elastic vibrations, through the opto-elastic effect[37]. In the conventional backscattering geometry, the phonon wavevector is $q = 4\pi n/\lambda$ in which $\lambda$ and $n$ are the laser excitation wavelength and the refractive index of the material at $\lambda$, respectively[38–40]. In our experiments, we used an excitation laser with $\lambda = 532$ nm. Figure 1h, i show the Brillouin spectra of pristine epoxy and a composite with 18.0 vol% of graphene loading, respectively. The data were accumulated at 7 K, 80 K, and RT at the same laser power on the sample surface. The peak observed at ~15 GHz at RT is attributed to the longitudinal acoustic (LA) phonons. The frequency of the LA phonon decreases with increasing temperature both in the pristine epoxy and composites as expected for the isotropic materials, which expand with the increasing temperature. We

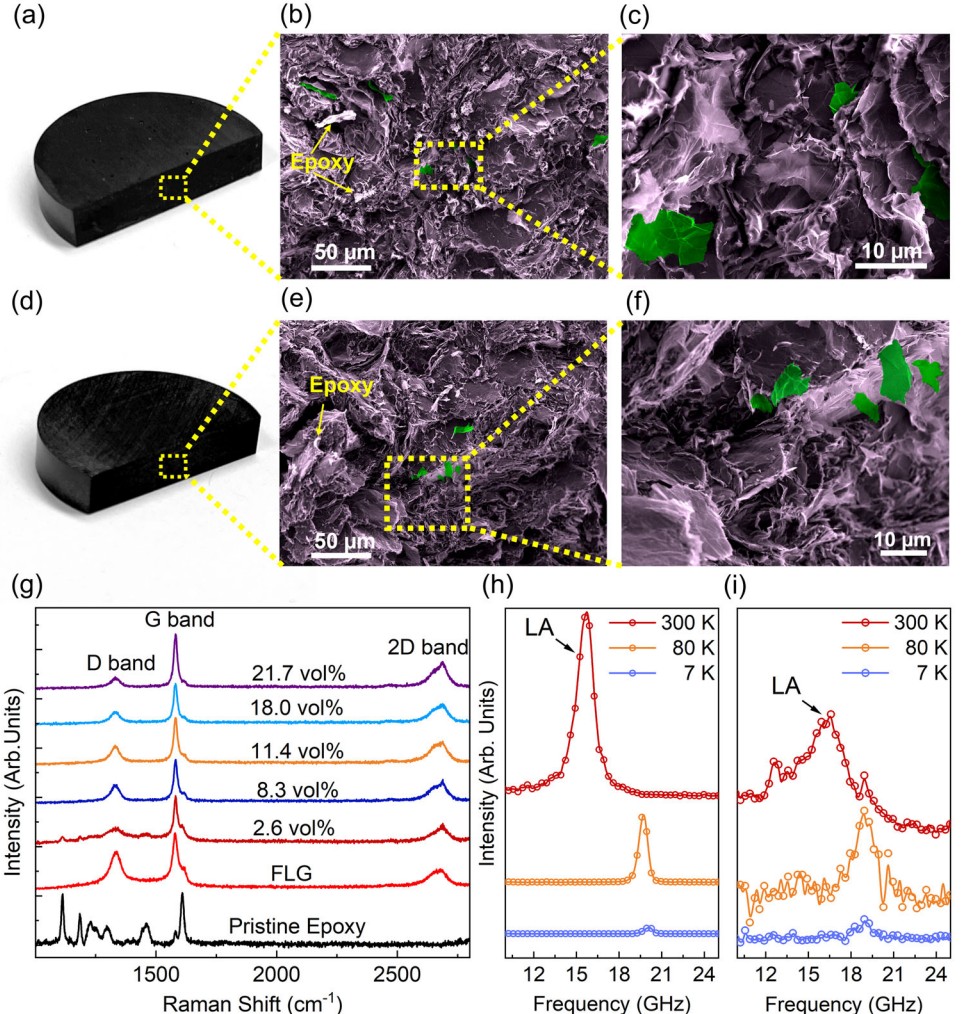

**Fig. 1 | Sample preparation and characterization.** From left to right, optical microscopy and cross-sectional SEM images of composites with (**a**–**c**) 2.6 vol% and, (**d**–**f**) 18.0 vol% loading of few-layer graphene. Pseudo-colors are used for clarity to illustrate the random distribution of fillers in the epoxy matrix. The green and violet regions represent graphene fillers and the epoxy matrix, respectively. **g** Room temperature Raman spectra of pristine epoxy, few-layer graphene, and composites with various graphene concentrations. Temperature-dependent Brillouin light scattering of **h** pristine epoxy, and **i** composite with 18.0 vol% of graphene filler loading. The peaks associated with the longitudinal acoustic phonons are denoted as LA.

have not observed signatures of transverse acoustic phonons, in agreement with the selection rules for the isotropic materials[38]. These observations provide additional support to the conclusion of well-dispersed and randomly oriented fillers in the composites. The data for the composites show more scatter owing to increased light absorption and decreased intensity of the scattered light. Similar trends in Brillouin signatures for the acoustic phonons in the pristine epoxy and epoxy with graphene indicate that the addition of the fillers has not negatively affected the mechanical properties of the material.

## Specific heat of graphene composites

We start with the investigation of the specific heat and illustrate its typical temperature trends in disordered amorphous materials. Figure 2a, b shows the results of the specific heat, $c_p$, measurements of pristine epoxy, and selected composites as a function of temperature for different filler concentrations. The data are presented in two different temperature ranges to make the trend more explicit (also, see Supplementary Fig. 4). The shaded area around the experimental data points indicates the standard error in the measurements conducted with Physical Property Measurement System (PPMS, DynaCool, Quantum Design, the USA). We used standard models to calculate the errors involved in our measurements. Other techniques such as

variance-based sensitivity analyses can also be used to quantify the influence of different parameters and assumptions on the measured values[41]. The details of the error analyses are described in the Supplementary Information. The results for the neat epoxy agree well with the literature[42]. The variation of the specific heat with temperature resembles that in other amorphous polymers[42–44]. In all samples, $c_p$ increases more rapidly in the low-temperature region; the rate of increase slows down as the temperature rises. To better visualize the temperature characteristics of $c_p$, we also plotted the data in a log-log scale, indicating the polynomial functional dependencies (Fig. 2c). One can distinguish three regions: quasi-cubic, $c_p \sim T^{3+\delta}$, in the temperature range of $2\,K \leq T \leq 6\,K$; parabolic, $c_p \sim T^2$, in the interval of $6\,K \leq T \leq 35\,K$; and linear, $c_p \sim T$, in the range of $T \geq 35\,K$. Among these regions, the $c_p$ behavior in the low-temperature limits is of particular interest since it deviates from the classical Debye model for crystalline materials. We plotted the "Debye-reduced" heat capacity, $c_p T^{-3}$, as a function of temperature in a log-log scale in Fig. 2d. In the insulating crystalline materials, $c_p \sim T^3$, and therefore one expects to see a flat curve in the low-temperature limits of the $c_p T^{-3}$ vs. $T$ plot. For the pristine epoxy and composites with graphene, there is no $T^3$ dependency of $c_p$. The latter stems from the amorphous, disordered natures of these materials.

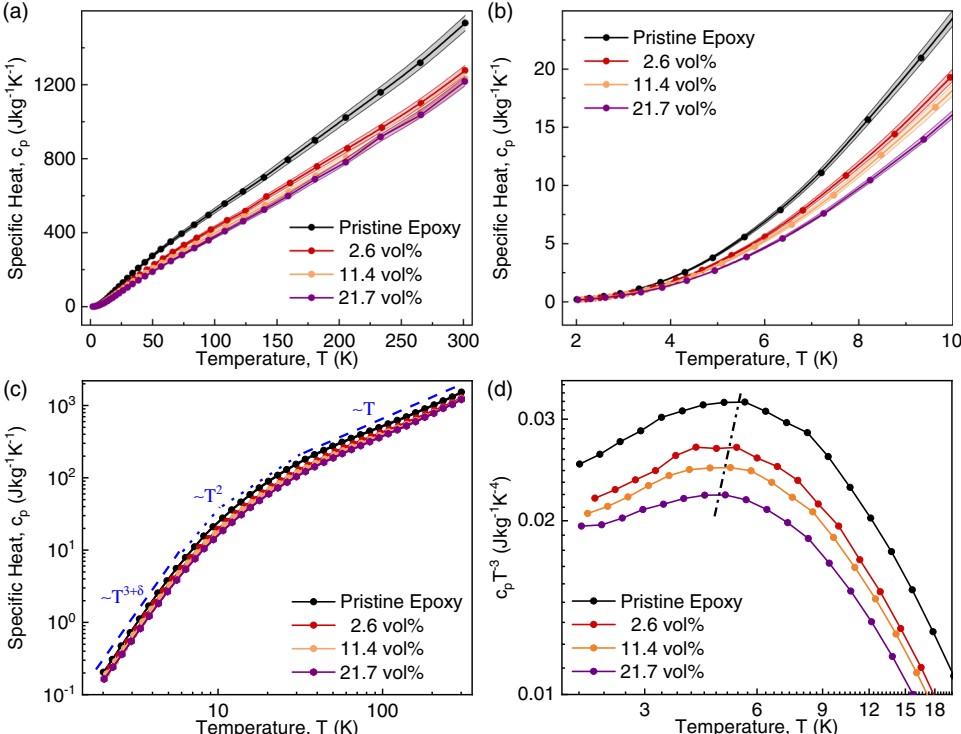

**Fig. 2 | Specific heat of graphene composites. a** Temperature-dependent specific heat of composites in the temperature range of $2\,\mathrm{K} \le T \le 300\,\mathrm{K}$. **b** The same data as in **a** is shown in the low-temperature limits. **c** Specific heat of the composites plotted in the log-log scale, revealing the quasi-cubic, parabolic, and linear temperature dependence in the low, intermediate, and high-temperature ranges. **d** The "Debye-reduced" specific heat of graphene composites as a function of temperature in a log–log scale. The dashed line is a guide to the eye, showing the behavior of the so-called "boson peak" as a function of filler loading and temperature.

One can notice from Fig. 2d that as the graphene loading increases, the curves flatten out, getting closer to the $c_p \sim T^3$ characteristic of crystalline materials. It is well-established that the specific heat of amorphous materials exhibits a "universal" characteristic in the low-temperature limits[45,46]. Below $T \sim 1\,\mathrm{K}$, the heat capacity considerably exceeds the Debye model predictions, dominated by a quasi-linear temperature dependence[45,46]. This behavior is explained by the tunneling model of two-level systems for amorphous materials at low temperatures[47]. This anomalous quasi-linear trend is followed by a hump in the $c_p T^{-3}$ data, referred to as the "boson peak" in the $3\,\mathrm{K} \le T \le 10\,\mathrm{K}$ region[47–49]. The "boson peak" shown in Fig. 2d is attributed to the low-frequency vibrational modes present in amorphous materials[49]. The temperature at which the boson peak occurs decreases slightly with increasing the filler loading. The "boson peak" eventually fades away with increasing the filler content as the material acquires more "crystalline" characteristics owing to the graphene content. The overall value of $c_p$ decreases with increasing graphene loading since $c_p$ of FLG is much lower than that of the epoxy matrix. We established the general trends of the specific heat of graphene composites that follow those of amorphous materials but become more crystalline-like with increasing graphene content.

## Thermal conductivity of graphene composites

We now turn to the main topic of this study—the thermal conductivity of graphene composites at cryogenic temperatures. Figure 3a shows the thermal conductivity, $k$, measured using PPMS in the temperature range from 2 K to RT in a log-log scale. The details of the measurements and error calculations are described in the Supplemental Information. The thermal conductivity of the pristine epoxy initially increases superlinearly in the temperature range of $2\,\mathrm{K} \le T \le 6\,\mathrm{K}$, followed by a "plateau region" in the interval of $6\,\mathrm{K} \le T \le 17\,\mathrm{K}$, where $k$ remains nearly constant. After the plateau, $k$ increases linearly again

until $T$-80 K at which point a second plateau extending up to $T$-175 K occurs. The existence of the first plateau in $k$ vs. $T$ dependence is universal for amorphous materials, and it occurs almost in the same temperature range. It starts at the temperature where the "boson peak" appears in the specific heat (see Fig. 2d)[50]. The first plateau region is explained by the two-level systems and the tunneling model[47,50]. According to this model, the dependence is due to the cross-over at which phonons with shorter MFP become dominant heat carriers instead of phonons with longer MFP[47,50]. The thermal conductivity is given by $k = (1/3)Cv\Lambda$ in which $C$, $v$, and $\Lambda$ are the specific heat, average group velocity, and the mean free path of phonons, respectively. In the plateau region, while $C$ increases with the temperature, $\Lambda$ decreases almost at the same rate. In other words, the plateau forms at the temperature range where the product of $\Lambda$ and $C$ becomes independent of temperature. The second plateau region was explained by similar considerations but its origin is still the subject of debate[51].

Before proceeding further with the analysis of the experimental thermal data we should clarify the use of the "phonon" concept and terminology. Naturally, the introduction of phonons—quanta of crystal lattice vibrations—requires translational symmetry encountered in crystalline solids[52,53]. Amorphous materials lack translation symmetry. Complex models involving different descriptions of atomic vibrations referred to as propagons, diffusons, and locons, corresponding to propagating, diffusion, and localized modes, have been introduced to describe thermal transport in amorphous materials[54,55]. However, in the context of the present study of heat conduction in epoxy-based composites, we still can use the concept of acoustic phonons as the elastic vibrations in the continuous medium of the base material. The acoustic phonons are the dominant heat carriers in dielectric materials, including amorphous epoxy. We can talk about the acoustic phonons, i.e. vibrations, propagating through the matrix that carry heat and interact with graphene fillers. For analysis of heat conduction in

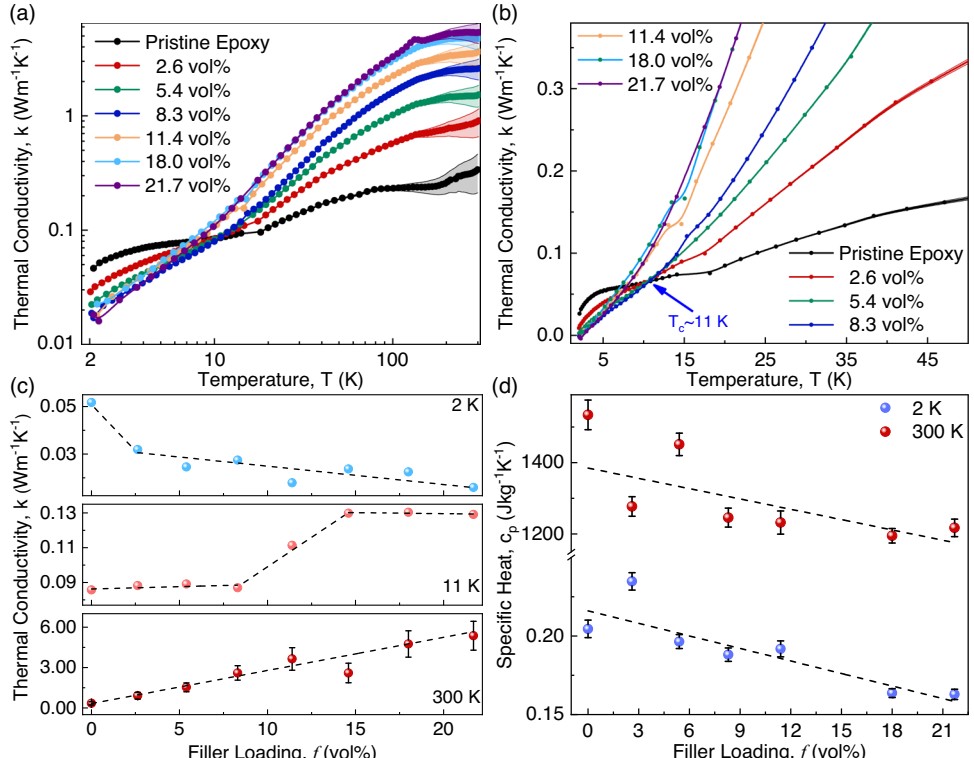

**Fig. 3 | Temperature-dependent thermal transport characteristics of graphene composites.** **a** Thermal conductivity of graphene composites in the temperature range of 2 K ≤ T ≤ 300 K in the log-log scale. The symbols are the experimental data points. The shaded area indicates the experimental uncertainty. **b** Thermal conductivity of the graphene composites in the low-temperature region, showing the cross-over temperature, $T_c$. **c** Thermal conductivity of the graphene composites as a function of the filler loading at 2 K, $T_c \sim 11$ K, and RT. The errors associated with the data for 2 K and 11 K are smaller than the size of the symbols and are not shown. **d** Specific heat of the graphene composites as a function of the filler loading at constant temperatures of 2 K and 300 K.

graphene composites, it is also useful to keep in mind the concept of the two-phase semi-crystalline medium where crystalline fillers interface with the amorphous polymer host[45].

Let us now consider the temperature-dependent characteristics of the thermal conductivity of the graphene–epoxy composites in more detail. In Fig. 3b, we replotted the thermal conductivity data for the same graphene loadings on a linear scale. There is no plateau region in graphene composites similar to the one observed for pristine epoxy. For composites with the filler loading $f \le 8.3$ vol%, there is a well-defined cross-over temperature, $T_c \sim 11$ K, at which $\kappa$ is the same for the pristine epoxy and graphene composites with variable graphene loading. The existence of a cross-over temperature and the absence of a plateau have been reported previously in the thermal conductivity data of semi-crystalline materials[56] and polymer-based composites with filler inclusions, which can be considered semi-crystalline media[45]. For example, while the plateau in $k$ emerges in the amorphous polyethylene terephthalate (PET), it vanishes in semi-crystalline PET containing crystalline zones with volume fraction as low as $f = 9$ vol%[56]. The $T_c \sim 15$ K is reported for PET with crystalline regions of $0 \le f \le 51$ vol% dispersed inside amorphous PET[56]. Several studies have reported a cross-over temperature for different composite systems in a similar temperature range. In all these studies, the plateau region vanished as soon as crystalline fillers were added to the amorphous polymer[28,57–60].

The peculiar features in the temperature-dependent thermal transport characteristics can be explained by the thermal boundary resistance (TBR), $r_b$, at the interface of the amorphous polymer and crystalline FLG fillers[28,56]. According to the acoustic mismatch theory, at sufficiently low temperatures, $r_b \sim T^{-3}$, and thereby $r_b$ shows strong effects in the low-temperature limits[61,62]. The effect becomes small at high temperatures. Note that, below $T_c$, $\kappa$ of the pristine epoxy is larger

than that of the epoxy composites with graphene fillers. This means that the inclusion of FLG fillers into the amorphous epoxy reduces its $k$ when the temperature is below $T_c$ but improves it when the temperature is above the cross-over temperature. This is a direct consequence of the strong temperature dependence of TBR at the polymer-filler interfaces. This will be further discussed in the theory section.

Figure 3c shows the thermal conductivity of graphene composites as a function of the filler loading, $f$, at constant temperatures of 2 K, $T_c = 11$ K, and RT. The systematic errors for the data points at low temperatures were smaller than the size of the symbols and not shown for clarity. At $T = 2$ K, the thermal conductivity of the epoxy falls approximately two times with the addition of only 2.6 vol% of graphene fillers. The sharp decrease in thermal conductivity suggests that FLG fillers contribute significantly to the phonon scattering processes. At this temperature, as the concentration of the filler increases, the thermal conductivity declines linearly. At RT, however, the trend is the opposite. The addition of graphene fillers improves the thermal conductivity monotonically in line with many prior reports[12,21,23]. The variation of $k$ as a function of $T$ is more intriguing in the vicinity of the cross-over temperature, $T_c \sim 11$ K. With increasing the graphene filler loading, thermal conductivity remains the same up to $f = 8.3$ vol%; then, linearly increases in the interval of 8.3 vol% $\le f \le 14.6$ vol%; and after that, remains unchanged with adding more fillers. Typically, a nonlinear behavior such as the one observed at $T_c \sim 11$ K might indicate that the composite system at $f = 14.6$ vol% enters the thermal percolation regime. The absence of such nonlinear behavior at $T = 2$ K and RT suggests an intriguing possibility of the temperature-dependent percolation threshold.

Typically, one thinks about the percolation threshold in terms of the filler loading, $f_H$, as the point where the fillers start to mechanically

touch each other, forming a continuous conductive network. Thermal percolation is less abrupt than electrical percolation because the heat can be conducted by the matrix materials, contrary to the electrical current that cannot be conducted by the dielectric matrix[21,63,64]. The ratio of the thermal conductivity of the fillers to the matrix, $k_f/k_m$, is several orders of magnitude smaller than the ratio of the electrical conductivity of the fillers to the matrix, $\sigma_f/\sigma_m$[21]. The wavelength of the thermal phonons, which make the dominant contribution to heat conduction, is $\lambda_T \sim hV_s/k_bT$, where $h$ is the Plank's constant, $k_b$ is the Boltzmann's constant and $V_s$ is the sound velocity, or more accurately, the phonon group velocity[45]. For a typical semiconductor material $\lambda_T$ is on the order of 1 nm−2 nm at RT[65]. This means that for the thermal percolation to occur at RT, the fillers should be in physical contact or close to each other, i.e., on the order of ~1 nm. At low temperatures, $\lambda_T$ increases by more than an order of magnitude. Thus, phonons with long wavelengths might provide thermal "cross-talk" to the fillers over some distance. The average distance between the fillers at ~10 vol % is still larger than $\lambda_T$. However, one should remember that more refined theories of thermal conductivity attribute a more significant contribution of phonons with wavelength above $\lambda_T$ to heat conduction[66]. If we accept this picture of the process, then the dependence of the thermal conductivity at 2 K and 300 K are those below and above thermal percolation, correspondingly (see Fig. 3c). It is illustrative to analyze them further with the specific heat dependence on the loading fraction shown in Fig. 3d. The specific heat decreases with increasing $f$ for both temperatures since $c_p$ of FLG is lower than that of the matrix. The thermal conductivity decreases at 2 K with $f$ either following the $c_p$ trend or because graphene fillers are acting more like scattering centers for the relevant low-wavelength phonons. The thermal conductivity at 300 K increases with $f$ despite the decrease owing to the addition of more percolated FLG conducting channels. The total cross-section of the percolated channels increases faster than the decrease in $c_p$.

It should be noted that the composites with graphene fillers can withstand cryogenic temperatures without degradation in their mechanical or thermal properties. The analyses of the optical microscopy images and results of thermal conductivity measurements of the composite with 5.4 vol% of graphene loading conducted during several temperature cycles between 2 K to 300 K confirm the mechanical and thermal stability of epoxy composites with FLG fillers at cryogenic temperatures (Supplementary Fig. 5). The optical microscopy image, taken from the surface of the same sample after three rounds of heating and cooling cycles, shows no micro crack development as a result of thermal cycling stresses.

## Effective medium model for the low-loading composites

We now develop a model to explain the heat conduction behavior in graphene composites. Different numerical methods and machine-learning approaches can be used to model the thermal conductivity in polymer-filler material systems[67,68]. In the present study, we use a model based on Nan's analytical effective medium approach to describe the underlying mechanisms contributing to the anomalous behavior observed in the thermal conduction of these composites in the low and high-temperature regions[69]. According to this model, the effective thermal conductivity of a composite with randomly oriented low-loading fillers is given as[69]

$$k = k_m \frac{3 + f[2\beta_{11}(1 - L_{11}) + \beta_{33}(1 - L_{33})]}{3 - f[2\beta_{11}L_{11} + \beta_{33}L_{33}]} \quad (1)$$

Here, $k_m$ is the thermal conductivity of the pristine epoxy, $k$ is the effective thermal conductivity of epoxy–graphene nanocomposite with the filler volume fraction $f$, $L_{ii}$ are the geometrical parameters that depend upon the aspect ratio, $p = t/L$, of graphene fillers with $t$ and $L$ being the thickness and lateral dimensions of the fillers. The details of

$L_{ii}$ parameters and their definition can be found in the Supplementary Information. The parameters $\beta_{ii}$ contain information about the thermal boundary resistance at the filler-epoxy interface and are defined as:

$$\beta_{ii} = \frac{K_{ii}^c - k_m}{k_m + L_{ii}(K_{ii}^c - k_m)} \quad (2)$$

where, $K_{ii}^c$ are the effective values of FLG thermal conductivity along different cartesian directions, that take into account the effect of interface thermal resistance. The effective thermal conductivities of the FLG fillers along the in-plane ($K_{11}^c$ - $K_{22}^c$) and through-plane ($K_{33}^c$) are, respectively,

$$K_{11}^c = K_{22}^c = \frac{k_{in}}{1 + \gamma L_{11} k_{in}/k_m} \quad (3)$$

$$K_{33}^c = \frac{k_{out}}{1 + \gamma L_{33} k_{out}/k_m} \quad (4)$$

where, $\gamma = (1 + 2p)\alpha$ in which $\alpha = r_b k_m/t$ is a dimensionless parameter related to the interface thermal resistance, $r_b$, between the epoxy and filler, $k_{in}$ and $k_{out}$ represent the in-plane and through-plane thermal conductivity of pristine graphene fillers, respectively. To obtain the temperature dependence of the effective thermal conductivity, all involved parameters in Eq. (1) including $k_{in}$, $k_{out}$, $k_m$, and $r_b$ were taken to be temperature-dependent (see Supplementary Information and Supplementary Fig. 6 and 7 for details). The combined effect of the high interfacial thermal resistance and low through-plane thermal conductivity of FLG at low temperatures (see Supplementary Fig. 6b and Fig. 7b) results in a significantly small in-plane and through-plane "effective thermal conductivity" of FLG, $K_{11}^c$ and $K_{33}^c$, computed using Eqs. (3) and (4). The results are shown in Fig. 4a. The effective through-plane thermal conductivity of FLG becomes lower than that of the neat epoxy through the whole temperature range of 2 K up to 300 K. The graphene fillers oriented perpendicular to the heat flux serve as the extra thermal boundary resistance, a scattering center, rather than the conduit of heat. Note that $K_{33}^c$ is more than two orders of magnitude lower than the thermal conductivity of the pristine epoxy at 2 K. The in-plane thermal conductivity of FLG is only three times higher than that of the pristine epoxy at $T = 2$ K, whereas, at RT, its effective in-plane thermal conductivity is ~160 times higher than that of the neat epoxy.

The results of the calculations based on this effective medium model for composites with $f \leq 8.3$ vol% are presented in Fig. 4b in the temperature range of 2 K up to 50 K. The model successfully reflects the experimental thermal conductivity characteristics for the low-concentration composites and correctly estimates the experimental cross-over temperature. Based on this model, we infer that the low effective through-plane thermal conductivity of FLG outweighs its highly effective in-plane thermal conductivity, causing the composite thermal conductivity to become lower than that of pure epoxy at low-temperature limits. As the loading of FLG increases, the effect of the low through-plane thermal conductivity of fillers dominates, resulting in decreasing the composite's thermal conductivity even more. In contrast, in high-temperature limits, both the in-plane and through-plane thermal conductivities increase, while simultaneously the interfacial TBR decreases as $r_b \sim T^{-3}$. This causes a rise in the effective thermal conductivity of the filler in both directions. At ~15 K, the effective thermal conductivities of FLG fillers recover enough, to result in a thermal transport enhancement. Above this temperature, the composite thermal conductivity becomes higher than that of the neat epoxy and it grows with increasing FLG loading. Thus, there is a transition in the thermal conductivity trend, with $k$ decreasing with increasing FLG concentration at low temperatures, and the opposite trend at higher temperatures. The latter leads to a cross-over temperature effect at 15 K for composite's $k$.

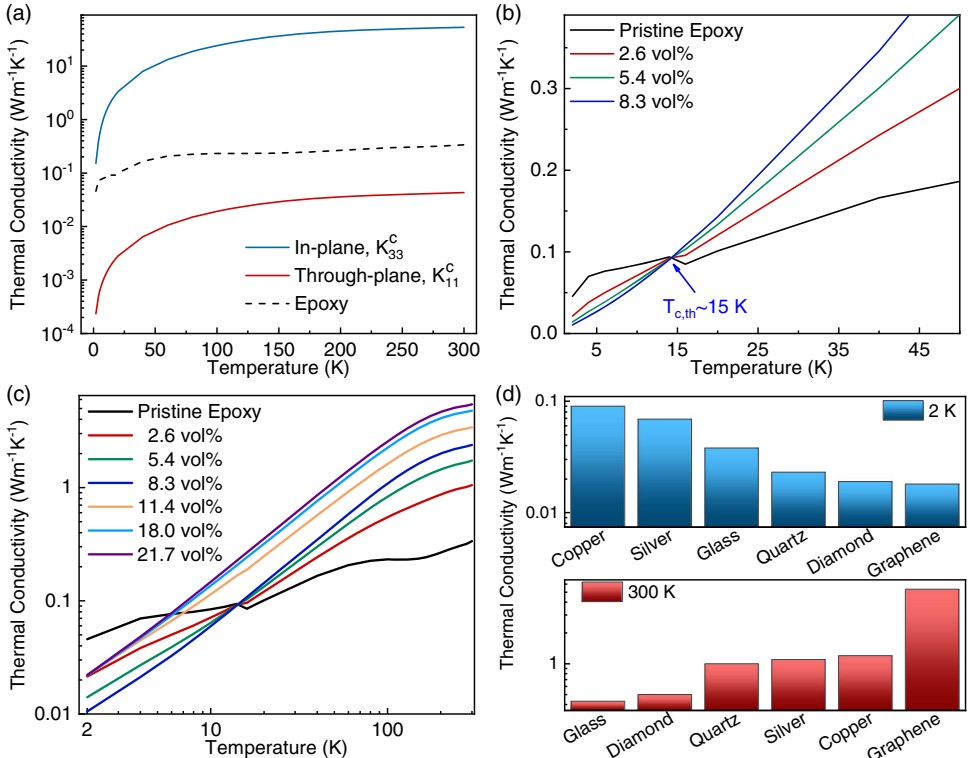

**Fig. 4 | Thermal conductivity and thermal performance benchmarking.**
**a** Calculated effective in-plane, $K_{33}^c$, and through-plane, $K_{11}^c$, thermal conductivity of few-layer graphene. **b** Calculated thermal conductivity of graphene composites with FLG loading of $f \leq 8.3$ vol%. The model successfully predicts the cross-over temperature, $T_{c,th} \sim 15$ K, which is in agreement with the experimental value. **c** Prediction of the percolative-based effective medium theory used to describe the thermal conductivity of composites with high filler loadings, $f \geq 11.4$ vol%. The data

is presented in the log-log scale together with the results of the low-loading model prediction presented in **b**. **d** Comparison of the measured heat conduction properties of graphene composites with other materials. Note that graphene composites demonstrate better thermal insulation at cryogenic temperatures and superior thermal conductivity at RT. The experimental data for composites with other fillers are from refs. [28,58].

## Effective medium model for the high-loading composites

The considered effective medium model is unable to predict the thermal conductivity of the high-concentration composites since it does not include the effect of filler-filler contact. As the filler loading increases, the probability that fillers can physically contact each other within the host polymer grows. Increasing the filler content can result in entering into a strong thermal percolation regime[21–23]. In this regime, heat can conduct along a network of highly conductive connected FLG fillers within the polymer matrix. The disappearance of the cross-over temperature in high-concentration samples implies that the favorable effect of heat conduction along such networks might have overcome the negative effect of high interfacial thermal resistance at low temperatures. To predict the thermal conductivity characteristics at higher FLG concentrations, we use a recently introduced percolation-based effective medium model[70]. Through this model the composite thermal conductivity $k$ is determined by solving the following equation:

$$(1-f)\frac{k_0 - k}{k + \frac{k_0 - k}{3}} + \frac{f}{3}\left[\frac{2(k_{11} - k)}{k + S_{11}(k_{11} - k)} + \frac{(k_{33} - k)}{k + S_{33}(k_{33} - k)}\right] = 0, \quad (5)$$

where $k_{11}$ and $k_{33}$ are the effective in-plane and through-plane thermal conductivity of FLG fillers, which account for the interfacial thermal resistance. These parameters are calculated using the actual in-plane and through-plane thermal conductivities of FLG, $k_{in}$ and $k_{out}$. The shape parameters $S_{11}$ and $S_{33}$ are related to the aspect ratio of the FLG fillers, and $k_0$ is the thermal conductivity of an interlayer surrounding FLG fillers. This interlayer represents the interface thermal resistance surrounding the graphene fillers. Its role is to include the combined

effect of interfacial thermal resistance at graphene-polymer and graphene-graphene contacts. The definitions of these parameters are explained in more detail in the Supplementary Information. The values of different parameters used in this model are listed in Supplementary Table 1. Figure 4c shows the results of the calculations for the high-concentration composites with $f \geq 11.4$ vol% along with the results of the low-loading effective medium model for the composites with $f \leq 8.3$ vol%. The predicted thermal conductivity in both cases is in good agreement with the experimental results (see Fig. 3). One can conclude that the superior heat conduction along the percolated graphene fillers channel overcomes the negative effect of large interfacial thermal resistance at low temperatures causing the cross-over effect to disappear at compositions greater than $f \geq 11.4$ vol%.

As follows from the above discussions, the thermal characteristics of semi-crystalline systems can drastically change in the low-temperature limits due to the increase in the dominant phonon wavelength and the changes in the phonon scattering mechanisms[28,45,56]. In this regime, fillers can act as phonon scattering centers rather than conductive inclusions, which suppresses the thermal transport of the composites even below the limit of its pristine amorphous polymer matrix[28,45,56]. The temperature-dependent thermal transport data of composites shows a cross-over temperature, $T_c$, usually in the interval of 5 K to 20 K, at which the thermal conductivity of the composite is lower than the pristine polymer host at temperatures below $T_c$ and vice versa. The inclusion of fillers causes the plateau region observed in amorphous polymers to disappear. These two features are attributed to TBR at the interface of polymer-filler, which becomes dominant at low temperatures[28]. Cryogenic characteristics of graphene composites are unique in the sense that they offer the strongest suppression of the

thermal conductivity below the cross-over temperature and the highest enhancement of the thermal conductivity above the cross-over temperature (see Fig. 4d). This is due to the atomic thickness of graphene and FLG, its geometry, and exceptionally high intrinsic in-plane thermal conductivity. For practical applications, graphene composites offer dual functionality for the circuits and systems where both cooling and thermal insulation are required. At RT, the thermal conductivity of the graphene composite with $f = 21.7$ vol% reaches $6\,\mathrm{Wm^{-1}K^{-1}}$, which is suitable for cooling semiconductor electronics, whereas a dilute composite with only 8.3 vol% filler loading reveals $k \sim 0.02\,\mathrm{Wm^{-1}K^{-1}}$ at $T = 2\,\mathrm{K}$, providing excellent thermal insulating properties for superconducting electronics.

In summary, we demonstrated that at cryogenic temperatures, the thermal conductivity of graphene composites can be both higher and lower than that of the reference pristine epoxy, depending on the graphene filler loading and specific temperature. There exists a well-defined cross-over temperature—above it, the thermal conductivity increases with the addition of graphene; below it, the thermal conductivity decreases with the addition of graphene. The counter-intuitive trend was explained by the specificity of heat conduction at low temperatures. The randomly distributed graphene fillers can serve, simultaneously, as the scattering centers for acoustic phonons in the matrix material and as the conduits of heat. We also argued that the onset of the thermal percolation threshold can undergo modification owing to the dominance of the low-wavelength phonons that facilitate the filler-to-filler heat conduction even before the fillers are physically connected. The obtained results suggest the possibility of using composites with the same constituent materials for, both, removing the heat and thermally insulating electronic components at cryogenic temperatures. The latter is an important capability for the development of quantum computing technologies and cryogenically cooled conventional semiconductor electronics.

## Methods

### Sample preparation and characterization
Several composite samples were prepared by mixing precalculated quantities of the epoxy resin, bisphenol-A (epichlorohydrin) with a molecular weight of 700 (Allied HighTech Products, Inc., the USA), the hardener, triethylenetetramine (Allied HighTech Products, Inc., the USA), and few-layer graphene (FLG) fillers (xGnP H-25, XG Sciences, the USA) to hit a targeted filler loading level. The average lateral dimension and surface area of the FLG fillers were 25 μm and $65\,\mathrm{m^2 g^{-1}}$, respectively. In order to have a uniform compound, FLG was added in several steps and mixed for 3 min at 800 rpm in a high-shear speed mixer (Flacktek, Inc., the USA). The hardener was then added to the epoxy resin at a mass ratio of 12:100. The final compound was mixed and vacuumed for 10 min to remove any possible trapped air bubbles. The latter was performed three times to achieve void-free composites. The samples were then poured into round silicon molds and left at room temperature for about 8 h to cure and solidify. At higher graphene concentrations, the samples were slightly pressed. Finally, all samples were heated at 130 °C in a furnace for 3 h. The final composite samples were disks with a diameter of 25.4 mm and a thickness of 5 mm. More details of the sample preparation and characterization are available in the Supplemental Information.

### Raman and Brillouin–Mandelstam light spectroscopy
Raman experiments were conducted in the conventional back-scattering configuration using a red laser with an excitation wavelength of 633 nm. All experiments were conducted at room temperature. The Brillouin–Mandelstam light scattering experiments were conducted in the backscattering configuration using a continuous-wave solid-state diode-pumped laser with an excitation

wavelength of 532 nm. Samples were launched in a specially designed helium-cooled stage. The temperature of the sample can be varied from 4 K up to 300 K. The laser light was focused using a ×10 objective with NA = 0.28. The laser power on the sample was kept low to avoid any possible laser-induced heating effects. This is important, especially for experiments at cryogenic temperatures. The scattered light was collected by the same objective and directed to the high-resolution 3 + 3 tandem Fabry-Perot interferometer (TFP-2, The Table Stable Ltd., Switzerland), detector, and spectrum analyzer.

### Heat capacity and thermal conductivity measurements
The heat capacity and thermal conductivity of the epoxy and composite samples were measured in the temperature range of 2 K to 300 K using a Quantum Design Physical Property Measurement System (PPMS). For thermal conductivity measurements, samples were cut into rectangular bars with $1 \times 1 \times 10\,\mathrm{mm^3}$ dimensions. The measurements were conducted using the steady-state 4-probe continuous mode. The heating rate was adjusted at $0.3\,\mathrm{K\,min^{-1}}$. The details of the heat capacity and thermal conductivity measurements as well as comprehensive error and uncertainty analyses are described in the Supplementary Information.

## Data availability
Any additional data is available from the authors upon request.

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

## Acknowledgements

A.A.B. acknowledges the support of the Vannevar Bush Faculty Fellowship from the Office of Secretary of Defense (OSD), under the Office of Naval Research (ONR) contract N00014-21-1-2947. A.A.B. and F.K. acknowledge the support of the National Science Foundation (NSF) via a Major Research Instrument (MRI) project DMR 2019056 entitled "Development of a Cryogenic Integrated Micro-Raman-Brillouin–Mandelstam Spectrometer."

## Author contributions

F.K. and A.A.B. conceived the idea of the low-temperature study of graphene composites for thermal management applications in cryogenic electronics and quantum computing technologies. F.K. and A.A.B. coordinated the project and contributed to the experimental data analysis. Z.E.N. exfoliated graphene fillers, prepared the graphene composites, performed SEM characterization, conducted Raman measurements, and contributed to the thermal data analysis. Y.X. conducted specific heat and thermal conductivity measurements using PPMS. D.W. conducted the Brillouin—Mandelstam light scattering measurements. J.O.B. assisted with the thermal data analysis. J.G. performed numerical modeling of thermal conductivity and contributed to data analysis. X.C. supervised thermal measurements and contributed to the experimental data analysis. F.K. and A.A.B. led the manuscript preparation. All authors contributed to the writing and editing of the manuscript.

## Competing interests

The authors declare no competing interests.
