## [Peer Review File · Nature Communications]

Cryogenic Characteristics of Graphene Composites – Evolution from Thermal Conductors to Thermal InsulatorsREVIEWER COMMENTS

Reviewer #1 (Remarks to the Author):

In this paper, the authors investigated the thermal properties of epoxy–graphene composites at temperatures from 2 K to room temperature (300 K). The results show the possibility of using composites with the same constituent materials for, both, removing the heat and thermally insulating electronic components at cryogenic temperatures. This is an impressive study that is well-constructed and nicely presented. I only have a few questions:

1. For practical applications, could the material withstand ultralow temperatures?
2. It is recommended to apply proof of concept or simulation to visualize the dual function of this material at different temperatures.
3. Overall, this study provides a new perspective on the design and application of thermal management materials.

Reviewer #2 (Remarks to the Author):

In this work, the authors demonstrated that at cryogenic temperatures, the thermal conductivity of graphene composites can be both higher and lower than that of the reference pristine epoxy, depending on the graphene filler loading and temperature. A critical temperature was observed, above it, the thermal conductivity of composites increases with the addition of graphene; below it, the thermal conductivity decreases with the addition of graphene. The underlying mechanism is the roles of graphene, as conductor and/or scattering center. The authors also propose a physical model that explains the experimental trends. This topic is interesting and the manuscript is well prepared. These results are very interesting, as well as provide a new insight in thermal management, if the mechanism can be justified. The following comments and suggestions should be considered before publication.

1. The impacts of graphene filler on thermal conductivity are really interesting, but the mechanism should be confirmed. In the proposed model, random orientation is assumed. Can the authors provide direct experimental results to demonstrate?
2. As thermal conductivity of graphene flake depends on its size, what is the role of flake size on this phenomenon? And the impact of phonon mean free path? How to explain the results from viewpoints of mean free path?
3. The recent advances on the thermal transport in two-dimensional materials [SCIENCE CHINA Physics, Mechanics & Astronomy, 65, 117002 (2022).] are highly related to this work and should be included to provide a timely survey of relevant literature studies for the readers.

Reviewer #3 (Remarks to the Author):

In this work, authors by conducting experimental tests show that at cryogenic temperatures, the thermal conductivity of graphene/epoxy composites can be both higher and lower than that of the reference epoxy polymer, depending on the graphene filler content and temperature. Authors also concluded that the graphene composites can be employed for, both, removing the heat and thermally insulating components at cryogenic temperatures. The study is original, well conducted and the obtained results are worthy of publication and such that I can recommend the publication of this manuscript provided that the authors address the following comments:

- 1- Why are the error bars not included in some of the measured data? How many samples were tested?
- 2- Another critical aspect about the thermal transport in the nanocomposites is related to the size and thickness of graphene fillers, I do not think that the simple analytical method can

capture these effects. Please include more details concerning the thickness and size distribution of graphene fillers.

3- Multi-layer graphene is a highly anisotropic material, how this nature can affect the modeling of thermal transport, some discussions can be useful.

4- Up to which temperature are the fabricated samples thermally stable? Can the graphene sheets improve the thermal stability of epoxy?

5- The manuscript would significantly benefit from a thorough theoretical or computational study that can explain underlying phenomena. At least authors should comment on approaches presented in Computational Mechanics, 2014, 53(5), 1047-1071 and Composites Science and Technology, 224, art. no. 109425 presenting such approaches including associated software.

6- Authors are also encouraged to include a detailed variance based sensitivity analysis as done for instance in Advances in Engineering Software, 2016, 100, 19-31 providing also a simple Matlab code.

REVIEWER COMMENTS

Reviewer #1 (Remarks to the Author):

In this paper, the authors investigated the thermal properties of epoxy–graphene composites at temperatures from 2 K to room temperature (300 K). The results show the possibility of using composites with the same constituent materials for, both, removing the heat and thermally insulating electronic components at cryogenic temperatures. This is an impressive study that is well-constructed and nicely presented. I only have a few questions:

Response: We appreciate Reviewer 1 for finding our results “impressive” and our manuscript “well-structured and nicely presented.” Below please find our point-by-point responses to your questions and comments:

1. For practical applications, could the material withstand ultralow temperatures?

Response: To address this comment, we measured the thermal conductivity of one composite sample with 5.4 vol% filler loading three times from 2 K to 300 K. We found that the thermal conductivity data are consistent, as shown in the figure below. In addition, the sample does not show any cracks after three cycles of measurements. These results confirm that our samples can withstand ultralow temperatures. We have included the new data in the Supplementary Information as Figure S5. We also edited the main text to accommodate the Reviewer’s suggestion.

Figure S5: (a) Thermal conductivity of the epoxy with 5.4 vol% graphene loading as a function of temperature cycling in the range of 2 K to RT. The inset shows an optical image of the sample after three measurements. No mechanical cracks were detected after several thermal cycling. (b) Thermal conductivity of the same sample shown in the cryogenic temperature range. The composite’s thermal conductivity does not exhibit any changes after three times of thermal cycling.

In the revised version, we added the following statement:

“It should be noted that the composites with graphene fillers can withstand the cryogenic temperatures without any degradation in their mechanical or thermal properties. The analyses of the optical microscopy images and results of thermal conductivity measurements of the composite with 5.4 vol% of graphene loading conducted during several temperature cycles between 2 K to 300 K confirms the mechanical and thermal stability of epoxy composites with FLG fillers at cryogenic temperatures (Figure S5). The optical microscopy image taken from the surface of the same sample after three rounds of heating and cooling cycles shows no micro crack development as a result of thermal cycling stresses.”

2. It is recommended to apply proof of concept or simulation to visualize the dual function of this material at different temperatures.

Response: We focused this study on the materials issues. The study includes both experimental and computational component to demonstrate the concept. We agree with the Reviewer that conducting proof of concept demonstration with an actual device at cryogenic temperature would be important for practical application. However, we feel that this task is beyond the scope of the present paper, and requires a separate project and a follow up publication. We used PPMS system for our cryogenic material characterization. The PPMS requires an alteration in its design in order to place inside a working prototype device to test the cryogenic thermal management. We started this re-design but it will take some time. For this reason, we reserve this practical application task for future publication. A description of the proof of concept experiment would be rather lengthy and require a separate paper. We hope the Reviewer would accept our explanation.

3. Overall, this study provides a new perspective on the design and application of thermal management materials.

Response: We thank Reviewer 1 for the positive consideration of our manuscript for publication in Nature Communications.

Reviewer #2 (Remarks to the Author):

In this work, the authors demonstrated that at cryogenic temperatures, the thermal conductivity of graphene composites can be both higher and lower than that of the reference pristine epoxy, depending on the graphene filler loading and temperature. A critical temperature was observed, above it, the thermal conductivity of composites increases with the addition of graphene; below it, the thermal conductivity decreases with the addition of graphene. The underlying mechanism is the roles of graphene, as conductor and/or scattering center. The authors also propose a physical model that explains the experimental trends. This topic is interesting and the manuscript is well prepared. These results are very interesting, as well as provide a new insight in thermal management, if the mechanism can be justified. The following comments and suggestions should be considered before publication.

Response: We thank Reviewer 2 for finding our results “interesting” with “new insight” into the thermal management of electronics. Below, please find our point-by-point response to your comments:

1. The impacts of graphene filler on thermal conductivity are really interesting, but the mechanism should be confirmed. In the proposed model, random orientation is assumed. Can the authors provide direct experimental results to demonstrate?

Response: We thank the Reviewer for an excellent question, which allowed us to further strengthen the manuscript. Per the reviewer’s suggestion, we carried out SEM characterizations of our samples with different filler loadings. In addition, we conducted Brillouin spectroscopy of the samples. To include the additional data, we revised our Figure 1. As seen in the revised Figure 1, graphene fillers are randomly distributed. Brillouin data further supports the conclusions about isotropic nature of the samples and random distribution of the fillers. We added the following paragraph.

“The composites were further characterized using Raman and Brillouin – Mandelstam light scattering spectroscopy, also referred to as the Brillouin light scattering spectroscopy. The Raman spectroscopy (Renishaw InVia) was carried out under the laser excitation with the wavelength of $\lambda = 633$ nm in the conventional backscattering configuration. In all experiments, the laser power was kept at ~ 3 mW. Raman spectra of several samples at random spots are presented in Figure 1(g). In all composites, the characteristic G-peak and 2D-peak of graphene were observed at ~ 1580 cm^{-1} and ~ 2700 cm^{-1} [Ref. ³⁰] confirming an even distribution of graphene throughout the samples. The intensity of the 2D peak is much lower than the G-peak indicating the presence of a mixture of single-layer and few-layer graphene in the composite samples. The Brillouin spectroscopy allows probing low-energy acoustic phonons near the Brillouin-zone (BZ) center with energies in the range of 2 GHz to 900 GHz.^{31–33} In polymers, the Brillouin spectrum is dominated by the inelastic scattering of light from bulk phonons, *i.e.* elastic vibrations, through the opto-elastic effect. In the conventional backscattering geometry, the phonon wavevector is $q = 4\pi n/\lambda$ in which λ and n are the laser excitation wavelength and the refractive index of material at λ , respectively.^{34–36} In our experiments, we used an excitation laser with $\lambda = 532$ nm. Figures 1 (h) and (i) show the Brillouin spectra of pristine epoxy and a composite with 18.0 vol% of graphene loading, respectively. The data were accumulated at 7 K, 80 K, and RT at the same laser power on the sample surface. The peak observed at ~ 15 GHz at RT is attributed to the longitudinal acoustic (LA) phonons. The frequency of the LA phonon decreases with increasing temperature both in the pristine epoxy and composites as expected for the isotropic materials, which expand with the increasing temperature. We have not observed signatures of transverse acoustic phonons, in agreement with the selection rules for the isotropic materials. These observations provide additional support to the conclusion of well-dispersed and randomly oriented fillers in the composites. The data for the composites shows more scatter owing to increased light absorption and decreased intensity of the scattered light. Similar trends in Brillouin signatures for the acoustic phonons in the pristine epoxy and epoxy with graphene indicate that addition of the fillers have not negatively affected mechanical properties of the material.”

2. As thermal conductivity of graphene flake depends on its size, what is the role of flake size on this phenomenon?

Response:

The Reviewer is correct that the thermal conductivity of the polymer composites depends on the lateral dimensions of the graphene fillers. We previously showed that at room temperature, the thermal conductivity of composites increases with increasing the size of the graphene fillers (please see: Sudhindra, S. *et al.* Specifics of Thermal Transport in Graphene Composites: Effect of Lateral Dimensions of Graphene Fillers. *ACS Appl. Mater. Interfaces* **13**, 53073–53082 (2021).) It was found that in the examined range of the lateral dimensions between 400 nm up to 1200 nm, the thermal conductivity of the composites increases with increasing size of the graphene fillers. The observed difference in thermal properties can be related to the average gray phonon mean free path (MFP) in graphene, which has been estimated to be around ~800 nm at room temperature. The thermal contact resistance of composites with graphene fillers of 1200 nm lateral dimensions was also smaller than that of composites with graphene fillers of 400 nm lateral dimensions.

Based on our prior experience, in this work, we intentionally used fillers with large lateral dimensions, larger than graphene's grey phonon mean free path which is ~800 nm at RT. This is essential to keep the intrinsic thermal conductivity of the fillers close to the intrinsic thermal conductivity of few layer-graphene. Naturally, the phonon MFP increases at low temperature. It is difficult to assess how the size of the fillers will affect the thermal conduction at cryogenics.

In general, the flake size would affect the overall thermal conductivity of the composite in two ways. (i) The flake size affects the intrinsic thermal conductivity of the graphene fillers. In this study, since we selected fillers with large lateral dimensions, it is unlikely that filler size settles the overall thermal conductivity characteristics. That is why in the analytical model, we used the temperature-dependent in-plane and through-plane thermal conductivity of few-layer graphene (Figure S6 a,b). (ii) The flake size would affect the overall thermal boundary resistance between the filler–polymer matrix. For two composites with similar filler concentrations and with fillers with two different average lateral dimensions, the effective thermal boundary resistance would be larger for the one with smaller filler size distribution due to the increased effective interface area between the fillers and the host polymer. Therefore, the thermal conductivity of the composite would be smaller for the one with smaller filler size. In the analytical model provided in this study, we used the temperature dependent thermal conductivity of graphite (Figure S6 a,b). The filler size effect comes into play in the parameters L_{ii} in the Nan's model and S_{ii} in the second model and determines the thermal boundary resistance or in other words, the effective in-plane and through-plane thermal conductivity of fillers. The smaller fillers would have smaller effective in-plane and through-plane thermal conductivities.

And the impact of phonon mean free path? How to explain the results from viewpoints of mean free path?

Response: The phonon mean free path is a strong function of temperature. At low temperatures, acoustic phonons with larger wavelength are the dominant heat carriers. In this case, graphene fillers act as the scattering centers and therefore, decrease the phonon MFP through point defect mechanism. As a consequence, the thermal conductivity of pristine epoxy is higher than that of the composites with graphene inclusions. At high temperatures, phonons with shorter wavelength contribute more to the heat

conduction. In this regime, crystalline fillers still act as scattering centers. However, at the same time, they act as conductive paths with orders of magnitude higher thermal conductivity compared to the polymer host. Therefore, the thermal conductivity of composites becomes higher than that of the pristine epoxy.

3. The recent advances on the thermal transport in two-dimensional materials [SCIENCE CHINA Physics, Mechanics & Astronomy, 65, 117002 (2022).] are highly related to this work and should be included to provide a timely survey of relevant literature studies for the readers.

Response: We thank the reviewer for suggesting this interesting paper. The reference has been cited in the revised manuscript.

Reviewer #3 (Remarks to the Author):

In this work, authors by conducting experimental tests show that at cryogenic temperatures, the thermal conductivity of graphene/epoxy composites can be both higher and lower than that of the reference epoxy polymer, depending on the graphene filler content and temperature. Authors also concluded that the graphene composites can be employed for, both, removing the heat and thermally insulating components at cryogenic temperatures. The study is original, well conducted and the obtained results are worthy of publication and such that I can recommend the publication of this manuscript provided that the authors address the following comments:

Response: We thank Reviewer 3 for finding our study “original, well-conducted” and recommending our manuscript for publication in Nature Communications. Below please find our point-by-point responses to your questions and comments.

1- Why are the error bars not included in some of the measured data? How many samples were tested?

Response: We assume the reviewer refers to the data shown on the two top panels of Figure 3 (b). In the temperature limits of 2 K and 11 K, the errors associated with the measurements fell within the size of the symbols and hence, we did not include them in the figure for clarity. In the original submitted manuscript, this was mentioned in the text and not in the caption. In the revised version, we added the explanation to the caption of Figure 3 as well. The full description of the error analyses is presented at the end of the manuscript and in the Supplementary Information.

2- Another critical aspect about the thermal transport in the nanocomposites is related to the size and thickness of graphene fillers, I do not think that the simple analytical method can capture these effects. Please include more details concerning the thickness and size distribution of graphene fillers.

Response: We agree with the Reviewer that no simple analytical model can capture all the effects on the thermal transport over the wide temperature range. We considered and tested several options. We concluded that the approach based on the analytical model of Nan, C. W., Birringer, R., Clarke, D. R. & Gleiter, H. Effective thermal conductivity of particulate composites with interfacial thermal resistance. J. Appl. Phys. 81, 6692 (1998) gave the best agreement with the experiment. The original model has been

successfully used to predict the thermal conductivity of polymer/graphene or polymer/carbon-nanotube composite leading to good agreement with experiments in prior studies. The model explicitly takes into account the particle size and thickness through the geometrical parameters L_{ii} , presented in Equation (1) in the main text of the manuscript. Specifically, the following lines in the manuscript discuss the dependence of L_{ii} on graphene dimensions:

“ L_{ii} are the geometrical parameters that depend upon the aspect ratio, $p = t/L$, of graphene fillers with t and L being the thickness and lateral dimensions of the fillers. The details of L_{ii} parameters and their definition can be found in the Supplementary Information.”

We further describe these geometrical parameters in the Supplementary information in equations S13 and S14. Below we discuss some of the works, where excellent agreement with experiments was obtained using Nan’s model.

1. The first work explicitly discusses the effect of graphene size on the thermal conductivity enhancement of polymer composites.

Michael Shtein, Roey Nativ, Matat Buzaglo, Keren Kahil, and Oren Regev, “Thermally Conductive Graphene-Polymer Composites: Size, Percolation, and Synergy Effects”, *Chemistry of Materials*, 27, 2100-2106 (2015).

The above work used Nan’s model (also used in our work) to achieve excellent agreement with experimental measurements. We refer the reviewer to Fig. 4a in the main manuscript of the above article for these results.

2. The second work listed below studies the effect of carbon nanotube aspect ratio on the thermal conductivity enhancement of carbon nanotube/polyethylene composites.

Tuba Evgin, Halil, Dogacan Koca, Nicolas, Horny, Alpaslan, Turgut, Ismail Hakkı Tavman, Mihai Chirtoc, Maria Omastová, Igor Novak, “Effect of aspect ratio on thermal conductivity of high-density polyethylene/multi-walled carbon nanotubes nanocomposites”, *Composites: Part A* 82, 208-213 (2016).

In this study, two different types of carbon nanotubes were used, one with an aspect ratio of around 500-3000, and the second with an aspect ratio of around 200-400. The thermal conductivity of polyethylene/carbon nanotube composites (prepared with the above two different types of nanotubes) was measured for a wide range of nanotube volume fractions. Results were compared against Nan’s model. The comparison showed that Nan’s model predictions matched experimental measurements for both types of nanotubes (with different sizes) over the entire volume fraction range studied. This comparison is shown in Fig. 4a of the above article. This work strongly suggests that Nan’s analytical model can accurately capture the effect of filler lateral size and thickness on the composite thermal conductivity.

Details concerning the thickness and size distribution of graphene fillers are included in the main text of the manuscript in the Experimental Section under the Materials subsection. Specifically, the following statement provides the details

“We used few-layer graphene with the specified average lateral dimension of 25 μm , an average thickness of 15 nm, and an average surface area of 50 to 80 m^2g^{-1} (xGnP, Grade H, XGSciences, the US) as the fillers for the preparation of the composites.”

3- Multi-layer graphene is a highly anisotropic material, how this nature can affect the modeling of thermal transport, some discussions can be useful.

Response: The anisotropic thermal conductivity of multi-layer graphene is explicitly included in the analytical model through the effective thermal conductivities K_{ii}^c along different cartesian directions. The expressions for K_{ii}^c are provided in Eqs. (3) and (4) in the main text. Specifically, we have provided the following discussion in the main text of the manuscript.

“ K_{ii}^c are the effective values of FLG thermal conductivity along different cartesian directions, that take into account the effect of interface thermal resistance. The effective thermal conductivities of the FLG fillers along the in-plane ($K_{11}^c \sim K_{22}^c$) and through-plane (K_{33}^c) are, respectively,

$$K_{11}^c = K_{22}^c = \frac{k_{in}}{1 + \gamma L_{11} k_{in} / k_m}, \quad (3)$$

$$K_{33}^c = \frac{k_{out}}{1 + \gamma L_{33} k_{out} / k_m}, \quad (4)$$

where, $\gamma = (1 + 2p)\alpha$ in which $\alpha = r_b k_m / t$ is a dimensionless parameter related to the interface thermal resistance, r_b , between the epoxy and filler, k_{in} and k_{out} represent the in-plane and through-plane thermal conductivity of pristine graphene fillers, respectively.”

The effect of anisotropy of graphene thermal conductivity is further discussed in the manuscript, where we show in Figure 4 (a) that the effective through-plane thermal conductivity of graphene becomes lower than the thermal conductivity of neat epoxy while the effective in-plane thermal conductivity stays higher than the neat epoxy, once the in-plane and out-of-plane thermal conductivities of pristine graphene are adjusted for the interfacial thermal resistance. As the temperature is lowered to the cryogenic regime, the *effective* through-plane thermal conductivity decreases dramatically (partly driven by an accompanying drop in interfacial thermal conductance as seen in Figure S7 (a) at lower temperatures) and becomes lower than neat epoxy by more than two orders of magnitude. Such graphene particles, when oriented perpendicular to the direction of heat transfer, act as thermal barriers rather than conductors of heat. It is this dramatic drop in effective through-plane thermal conductivity of graphene that causes the composite thermal conductivity to become lower than neat epoxy values at cryogenic temperatures.

The following lines in the manuscript relate to the above discussion:

“The combined effect of the high interfacial thermal resistance and low through-plane thermal conductivity of FLG at low temperatures (see Figure S6 (b) and Figure S7 (b)) results in a significantly small in-plane and through-plane “effective thermal conductivity” of FLG, K_{11}^c and K_{33}^c , computed using equation (3) and (4). The results are shown in Figure 4 (a). The effective through-plane thermal conductivity of FLG becomes lower than that of the neat epoxy through the whole temperature range of 2 K up to 300 K. The graphene fillers oriented perpendicular to the heat flux serve as the extra thermal boundary resistance, a scattering center, rather than the conduit of heat. Note that K_{33}^c is more than two orders of magnitude lower than the thermal conductivity of the pristine epoxy at 2 K. The in-plane thermal conductivity of FLG is only three times higher than that of the pristine epoxy at $T = 2$ K, whereas, at RT, its effective in-plane thermal conductivity is ~ 160 times higher than that of the neat epoxy.”

4- Up to which temperature are the fabricated samples thermally stable? Can the graphene sheets improve the thermal stability of epoxy?

Response: We have examined the stability and thermal and electromagnetic interference shielding of graphene-based composites in some of our prior studies. For example, in “Barani, Z. et al. Multifunctional Graphene Composites for Electromagnetic Shielding and Thermal Management at Elevated Temperatures. *Adv Electron Mater* 6, 2000520 (2020)” it was shown that the electromagnetic interference shielding properties of the composites enhances at elevated temperatures up to 520 K. The samples were not degraded after multiple experiments. The thermal conductivity properties of epoxy with graphene and epoxy with copper and graphene fillers have been examined and reported by us in “Kargar, F. et al. Thermal percolation threshold and thermal properties of composites with high loading of graphene and boron nitride fillers. *ACS Appl Mater Interfaces* 10, 37555–37565 (2018)” and “Barani, Z. et al. Thermal Properties of the Binary-Filler Hybrid Composites with Graphene and Copper Nanoparticles. *Adv Funct Mater* 30, 1904008 (2020).” In both these studies, samples were stable in the examined temperature ranges (up to 450 K). In this study, we focused on the temperature-dependent thermal conductivity of the samples in the cryogenic and RT range. Per reviewer’s suggestion, we added the results of thermal conductivity measurements of the composite with 5.4 vol% graphene loading during thermal cycling from 2 K to 300 K. The results are presented in Figure S5. No thermal or mechanical degradation was observed in the sample.

Figure S5: (a) Thermal conductivity of the epoxy with 5.4 vol% graphene loading as a function of temperature cycling in the range of 2 K to RT. The inset shows an optical image of the sample after three measurements. No mechanical cracks were detected after several thermal cycling. (b) Thermal conductivity of the same sample shown in the cryogenic temperature range. As seen, the composite's thermal conductivity does not exhibit any changes after three times of thermal cycling.

5- The manuscript would significantly benefit from a thorough theoretical or computational study that can explain underlying phenomena. At least authors should comment on approaches presented in Computational Mechanics, 2014, 53(5), 1047-1071 and Composites Science and Technology, 224, art. no. 109425 presenting such approaches including associated software.

Response: We thank the reviewer for bringing these works to our attention. The approach based on machine learning to predict material properties is impressive and useful for a wide range of problems.

The machine learning-based approach outlined in the above works uses a finite element model to simulate the effect of carbon nanotubes on composite thermal conductivity. A lot of the parameters used in such an approach (when applied to our system) such as the graphene thermal conductivity, interfacial thermal resistance, etc. would be the same as those used in our analytical model. Simulating particle-particle contact may pose some issues related to mesh generation. While such analysis is beyond the scope of the present study, we will consider it for future work. We have referenced the works discussed by the Reviewer in our manuscript.

6- Authors are also encouraged to include a detailed variance based sensitivity analysis as done for instance in Advances in Engineering Software, 2016, 100, 19-31 providing also a simple Matlab code.

Response: We went through the recommended paper and agree that a variance-based sensitivity analyses is a promising approach. We cited this interesting paper and will use the recommended MATLAB tool in our future studies.

REVIEWERS' COMMENTS

Reviewer #1 (Remarks to the Author):

It is suitable for publication now.

Reviewer #2 (Remarks to the Author):

The authors addressed the reviewer's comment and suggestion. The manuscript is acceptable now.

Reviewer #3 (Remarks to the Author):

I have no more comments.